# Deeper response predicts better outcomes in high-risk-smoldering-myeloma: results of the I-PRISM phase II clinical trial

Omar Nadeem[1,2,8] ✉, Michelle P. Aranha [1,2,3,8], Robert Redd [4], Michael Timonian [1,2,3], Sophie Magidson [1], Elizabeth D. Lightbody[1,2,3], Jean-Baptiste Alberge [1,2,3], Luca Bertamini[1,2,3], Ankit K. Dutta[1,2,3], Habib El-Khoury[1,2,3], Mark Bustoros [1,5], Jacob P. Laubach[1], Giada Bianchi[6], Elizabeth O'Donnell[1], Ting Wu[3], Junko Tsuji[3], Kenneth C. Anderson [1], Gad Getz[2,3,7], Lorenzo Trippa[4], Paul G. Richardson[1], Romanos Sklavenitis-Pistofidis[1,2,3] & Irene M. Ghobrial [1,2,3] ✉

Early therapeutic intervention in high-risk smoldering multiple myeloma (HR-SMM) has shown benefits, however, no studies have assessed whether biochemical progression or response depth predicts long-term outcomes. The single-arm I-PRISM phase II trial (NCT02916771) evaluated ixazomib, lenalidomide, and dexamethasone in 55 patients with HR-SMM. The primary endpoint, median progression-free survival (PFS), was not reached (NR) (95% CI: 57.7–NR, median follow-up 50 months). The secondary endpoint, biochemical PFS, was 48.6 months (95% CI: 39.9–NR) and coincided with or preceded SLiM-CRAB in eight patients. For additional secondary objectives, the overall response rate was 93% with 31% achieving complete response (CR) and 45% very good partial response (VGPR) or better. CR correlated strongly with the absence of SLiM-CRAB and biochemical progression. MRD-negativity ($10^{-5}$ sensitivity) predicted a 5-year biochemical PFS of 100% versus 40% in MRD-positive patients (p = 0.051), demonstrating that deep responses significantly improve time to progression. Exploratory single-cell RNA sequencing linked tumor MHC class I expression to proteasome inhibitor response, and a lower proportion of GZMB+ T cells within clonally expanded CD8+ T cells associated with suboptimal outcomes.

Smoldering multiple myeloma (SMM) is a heterogenous disease, with an average risk of progression to myeloma (MM) of 10% per year[1]. Patients with ~50% risk of developing overt MM (myeloma associated organ damage) over 2 years are considered high-risk SMM (HR-SMM)[2].

Several risk models[3,4] have been developed to define this group, including the Mayo 2008 criteria which define HR-SMM as having ≥10% plasmacytosis, M protein ≥3 g/dl, and serum free light chain (SFLC) ratio ≥8 or ≤0·125. In 2015, Rajkumar et al. compiled features to define

[1]Center for Early Detection and Interception of Blood Cancers, Department of Medical Oncology, Dana-Farber Cancer Institute, Boston, MA, USA. [2]Harvard Medical School, Boston, MA, USA. [3]Broad Institute of MIT and Harvard, Boston, MA, USA. [4]Department of Biostatistics and Computational Biology, Dana-Farber Cancer Institute, Boston, MA, USA. [5]Division of Hematology and Medical Oncology, Meyer Cancer Center, New York-Presbyterian Hospital, New York, NY, USA. [6]Amyloidosis Program, Division of Hematology, Brigham and Women's Hospital and Dana Farber Cancer Institute, Boston, MA, USA. [7]Cancer Center and Department of Pathology, Massachusetts General Hospital, Boston, MA, USA. [8]These authors contributed equally: Omar Nadeem, Michelle P. Aranha. ✉e-mail: omar_nadeem@dfci.harvard.edu; irene_ghobrial@dfci.harvard.edu

HR-SMM[5]. Later the International Myeloma Working Group (IMWG) introduced the 20/2/20 criteria for high-risk classification, identifying patients with at least two of the following: bone marrow plasmacytosis >20%, M-protein >2 g/dL, and SFLC ratio >20[6,7].

SMM management has historically consisted of observation. However, advances in risk stratification and MM therapy suggest that early therapeutic intervention may improve prognosis and prevent MM morbidity. Two phase III trials of lenalidomide or lenalidomide and dexamethasone in patients with HR-SMM demonstrated improvement in progression-free survival (PFS)[8,9] with one study showing an improvement in overall survival, supporting early intervention[10]. While the prolongation of overall survival with early intervention is a subject of debate, extending PFS may delay potentially irreversible end organ damage that severely impact patients' quality of life[11–13].

Despite multiple phase II trials evaluating triplet or quadruplet therapies for HR-SMM[14–17], there are many unresolved questions regarding optimal management of patients with HR-SMM, especially in adapting response metrics from overt MM such as depth of response, minimal residual disease (MRD), and biochemical progression for HR-SMM. The studies using lenalidomide with or without dexamethasone had low rates of complete response (CR) and therefore could not assess the role of depth of response in long-term follow-up.

The I-PRISM trial evaluated the combination of ixazomib, lenalidomide, and dexamethasone (IxaRD) in HR-SMM. We addressed several critical questions for therapeutic intervention in HR-SMM, including whether (1) depth of response (MRD negativity, CR, or VGPR) impacts the long-term outcome of patients treated with a lenalidomide-based therapy, (2) biochemical progression correlates with end-organ damage, (3) 20/2/20 defines participants most likely to benefit from early intervention with lenalidomide-based therapy, and (4) genomic biomarkers could refine risk stratification of participants enrolled on these studies.

## Results

### Baseline characteristics

Between March 2017 and February 2020, 55 patients were enrolled on this study (Fig. 1A). Patient baseline characteristics, definition of HR-SMM, and high-risk cytogenetics are summarized in Table 1A and Supplementary Table 1. The median age was 64 (range 40–84) with 30 (55%) males. Patient eligibility was determined using the Rajkumar et al. high-risk criteria described in the eligibility criteria in "*Methods*"[5]. For cross-study comparisons, we present the risk stratification obtained using several validated high-risk models (Table 1B), including the Mayo 2008, Mayo 2018, IMWG risk score tool, and high-risk cytogenetics. High-risk FISH markers were gain1q, del13q, del17p, del1p, t(4;14), or t(14;16).

While FISH testing is a standard clinical test for prognostication, it can fail to return results due to too few cells. 36% of FISH analyses failed (FISH at screening was successful in 35/55; Fig. 1B), so we employed whole genome sequencing (WGS) and single-cell RNA sequencing (scRNA-seq) to identify high-risk translocations and copy number alterations (CNAs; Fig. 1B). Expression of primary IgH translocation marker genes from scRNA-seq data was used to infer the presence of translocations. The expression of relevant marker genes was highly concordant with FISH results, and we detected additional translocations in three patients for whom FISH had failed due to low cell number. Furthermore, we inferred CNAs from scRNA-seq data, detecting hyperdiploidy in another four patients, and additional high-risk secondary CNVs such as amp1q, del13q, and del1p in four patients. WGS confirmed the IgH translocations, hyperdiploidy, and secondary CNVs in 10 patients with both WGS and scRNA-seq of tumor cells. No discrepancies were observed between scRNA-seq and WGS. Overall, for ten patients where FISH failed, scRNA-seq and WGS data existed, and we were able to resolve the cytogenetic abnormalities for these cases. Further, by reviewing pre-screening FISH reports, we obtained cytogenetic data for 49 of 55 individuals. Since we did not have whole-exome or WGS data on all patients, we refrained from assessing the impact on outcomes of certain high-risk mutations, including MAPK mutations, DNA repair mutations, or MYC alterations, which have been previously identified as high risks[18–20]. Recurrent mutations in oncogenic drivers and CNVs detected by WGS are shown in Supplementary Fig. 1.

### Safety

Adverse events are detailed in Table 2. The most common treatment-related toxicities of any grade included leukopenia in 44 patients (80%), neutropenia in 43 patients (78%), fatigue in 42 patients (76%), rash in 38 patients (69%), and diarrhea in 37 patients (67%). One patient developed a grade 5 intracranial hemorrhage 3 weeks after completion of therapy that was not related to the study intervention; the patient was on therapeutic anticoagulation with a history of atrial fibrillation and the event was considered unrelated by the patient's treating physicians. Five patients developed non-hematological malignancies while on follow-up (one instance each of prostate, head/neck, ovarian, uterine, and melanoma). Of critical importance, no patients were diagnosed with secondary myeloid malignancies or myelodysplastic syndrome. Details regarding dose modifications and treatment discontinuations are provided in Supplementary Data.

### Efficacy

The overall response rate (partial response [PR] or better according to IMWG criteria[21]) was 93% (*n* = 51), with CR observed in 17 patients (31%), VGPR in 8 patients (15%), and PR in 26 patients (47%; Table 3). All patients who achieved CR also achieved a stringent and sustained CR for at least 6 months with responses deepening over time (Fig. 2). Twenty-five patients (45%) achieved VGPR or better and 54 (98%) achieved minimal response or better. The overall response rate was similar for patients in each 20/2/20 risk groups: low (100%), intermediate (90%), and high (92%; Cochran-Armitage test, *p* = 0.55).

Eight participants progressed to overt MM (SLiM-CRAB criteria)[22] (Fig. 2), and they all experienced biochemical progression concurrent with or before SLiM-CRAB progression (Fig. 2). One patient discontinued protocol therapy after one cycle due to unrelated comorbid conditions, including colitis and atrial fibrillation, and remained on follow-up for 10 months before SLiM-CRAB progression. The remaining seven progressors completed all twenty-four cycles of protocol therapy and continued on follow-up before progression. Four of eight progressors had a SLiM-CRAB event within 1 year of their final treatment, and the median time to progression for the remaining four progressors was 21 months (range: 13.2–32.7 months). The median time to progression for all 8 patients who progressed with SLiM-CRAB criteria was 23 months (range 2.0–32.7 months). The SLiM-CRAB events were as follows: four patients developed lytic bone disease noted on PET or MRI during follow-up, one patient developed renal failure and hypercalcemia 9 months after completion of therapy (after showing biochemical progression at 6 months post-therapy), one patient developed anemia 17 months post-therapy after showing biochemical progression at 6 months post-therapy, one patient met SLiM-CRAB criteria at 13 months post-therapy with FLC ratio >100 and BMPCs of 60% prompting initiation of therapy, and one patient met SLiM-CRAB criteria at 13 months post-therapy with FLC ratio >100. Seven of the eight patients who developed SLiM-CRAB had at least one high-risk genetic event at baseline [1q gain (*n* = 4), del13q (*n* = 5), del17p (*n* = 2, one confirmed from FISH tests and one identified in a pre-screening FISH report from patient records), del1p (*n* = 3), t(14;16) (*n* = 1), KRAS (*n* = 2), TP53 (*n* = 1)] (Fig. 1B). Among all progressors, the best response achieved was PR.

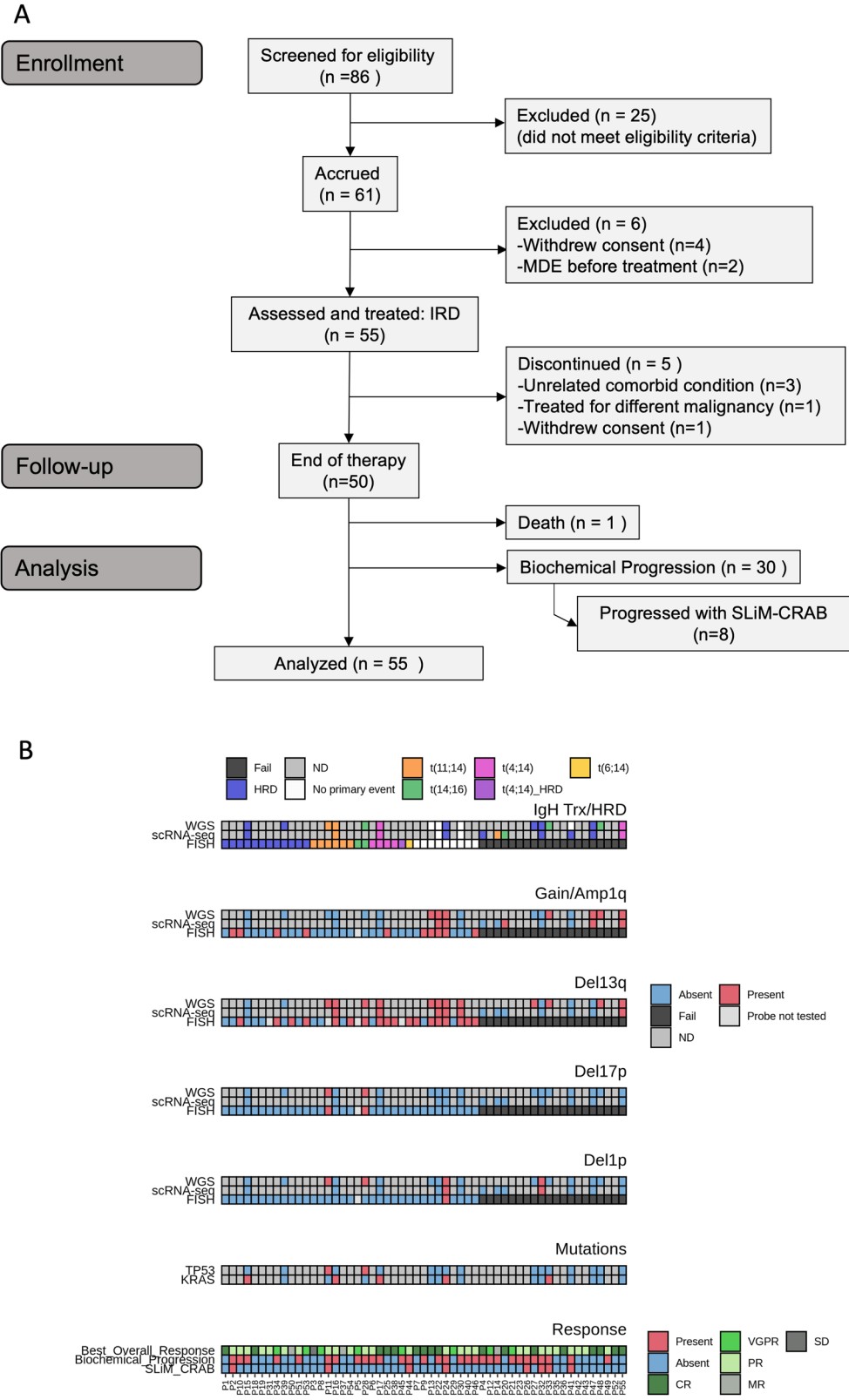

**Fig. 1 | Study design, patients' cytogenetic abnormalities, and responses.**
**A** CONSORT diagram. **B** Primary events such as translocations and hyperdiploidy and secondary events such as copy number alterations identified by FISH, WGS, scRNA, and response to therapy. WGS whole genome sequencing, scRNA single-cell RNA sequencing. scRNA-seq revealed a more complex aberration in Patient 41 (HRD with trisomies of chromosomes 5, 9, 11, 15, 19, and 21) compared to WGS which only detected trisomies on chromosomes 9 and 15. This explains the apparent discrepancy between WGS finding "no primary event" and scRNA-seq identifying HRD (homologous recombination deficiency) in that case. ND not determined, CR complete response, VGPR very good partial response, PR partial response, MR minimal response, SD stable disease. Source data are provided as a Source Data file.

**Table 1 | (A) Baseline characteristics (B) Baseline risk stratification**

| A | (N = 55) |
|---|---|
| Median age—year (range) | 64 (40–84) |
| Male sex—no. (%) | 30 (55%) |
| Race—no./N (%) | |
| White/Caucasian | 53 (96%) |
| Asian | 1 (2%) |
| Other | 1 (2%) |
| Ethnicity—no./N (%) | |
| Hispanic | 2 (4%) |
| Non-Hispanic | 53 (96%) |
| ECOG performance status—no./N (%) | |
| 0 | 46 (84%) |
| 1 | 9 (16%) |
| Smoldering myeloma disease type—no./N (%) | |
| IgG | 37 (67%) |
| IgA | 16 (29%) |
| Light chain only | 2 (4%) |
| Evolving Disease—no./N (%) | |
| Yes | 29 (53%) |
| No | 26 (47%) |
| Cytogenetic risk category—no./N (%) | |
| High-risk [t(4;14), t(14;16), del(17p), 1q gain, del1p, del13q | 35 (64%) |
| Standard-risk | 14 (25%) |
| Not determined | 6 (11%) |
| [a]Cytogenetic abnormalities—no./N (%) | |
| t(4;14) | 6 (11%) |
| t(14;16) | 6 (11%) |
| del(17p) | 3 (5%) |
| 1q gain/amp | 18 (33%) |
| t(11;14) | 8 (15%) |
| Monosomy 13 | 26 (47%) |
| 20/2/20 Risk | |
| High | 25 (45%) |
| Intermediate | 21 (38%) |
| Low | 9 (16%) |
| Mayo 2008 | |
| 1 risk factor | 13 (24%) |

**Table 1 (continued) | (A) Baseline characteristics (B) Baseline risk stratification**

**A**

| | (N = 55) |
|---|---|
| 2 risk factors | 40 (72%) |
| 3 risk factors | 2 (4%) |

**B**

| Patient ID | IgA Isotype[b] | M-spike ≥ 3.0[c] | Immunoparesis[d] | FLC Ratio ≥ 8[e] | Evolving Disease[f] | BMPC 50–60%[g] | t(4;14) -17p +1q[h] | High-Risk cytogenetics FISH[i] | High-Risk cytogenetics FISH+Seq[j] | Mayo 2008 | IMWG 20/2/20 | IMWG High Precision | Number of High-Risk Criteria[k] | Total number of eligibility criteria met (+BMPC ≥ 10%)[l] |
|---|---|---|---|---|---|---|---|---|---|---|---|---|---|---|
| P1 | 1 | 0 | 0 | 1 | 1 | 0 | 0 | 0 | 0 | Int | Int | Low-Int | 1 | 3 |
| P2 | 0 | 1 | 1 | 1 | 1 | 1 | 1 | 1 | 1 | High | High | High | 5 | 6 |
| P3 | 1 | 0 | 1 | 0 | 0 | 0 | 0 | 0 | 0 | Low | Int | Low | 0 | 2 |
| P4 | 0 | 0 | 1 | 1 | 1 | 0 | N/A | N/A | 0 | Int | Low | Low | 1 | 3 |
| P5 | 0 | 0 | 0 | 1 | 1 | 0 | 0 | 1 | 1 | Int | High | High | 4 | 2 |
| P6 | 1 | 0 | 0 | 0 | 0 | 0 | 1 | 1 | 1 | Low | Low | Low | 1 | 2 |
| P7 | 1 | 0 | 1 | 0 | 0 | 0 | 0 | 1 | 1 | Low | Int | Low-Int | 1 | 2 |
| P8 | 1 | 0 | 1 | 0 | 0 | 0 | 0 | 0 | 0 | Low | Low | Low | 0 | 2 |
| P9 | 0 | 0 | 1 | 1 | 0 | 0 | 0 | 0 | 1 | Int | High | Low-Int | 2 | 2 |
| P10 | 0 | 0 | 1 | 1 | 0 | 1 | 1 | 1 | 1 | Int | Int | High | 2 | 4 |
| P11 | 0 | 0 | 1 | 1 | 1 | 0 | 1 | 1 | 1 | Int | Int | Int | 2 | 4 |
| P12 | 1 | 0 | 0 | 0 | 1 | 0 | 1 | 1 | 1 | Low | Low | Low | 2 | 3 |
| P13 | 0 | 0 | 1 | 1 | 0 | 0 | 1 | 1 | 1 | Int | Low | Int | 1 | 3 |
| P14 | 0 | 0 | 1 | 0 | 0 | 0 | N/A | N/A | N/A | Low | High | Low-Int | 1 | 1 |
| P15 | 0 | 1 | 0 | 0 | 1 | 1 | 0 | 0 | 0 | Int | High | Int | 2 | 3 |
| P16 | 0 | 0 | 0 | 1 | 1 | 0 | 0 | 1 | 1 | Int | High | Int | 3 | 2 |
| P17 | 0 | 1 | 0 | 1 | 0 | 1 | 1 | 1 | 1 | High | High | High | 4 | 4 |
| P18 | 0 | 0 | 1 | 1 | 1 | 0 | 0 | 0 | 0 | Int | High | High | 2 | 2 |
| P19 | 1 | 0 | 1 | 1 | 0 | 0 | 0 | 0 | 0 | Int | Int | Int | 0 | 3 |
| P20 | 0 | 0 | 0 | 1 | 0 | 0 | N/A | N/A | 1 | Int | Int | Low-Int | 1 | 1 |
| P21 | 1 | 0 | 0 | 0 | 1 | 0 | N/A | N/A | N/A | Low | Low | Low | 1 | 2 |
| P22 | 0 | 0 | 1 | 1 | 0 | 0 | 1 | 1 | 1 | Int | High | High | 3 | 2 |
| P23 | 0 | 0 | 0 | 1 | 1 | 0 | N/A | N/A | N/A | Int | Low | Low | 1 | 2 |
| P24 | 0 | 0 | 1 | 1 | 1 | 1 | 1 | 1 | 1 | Int | High | High | 4 | 5 |
| P25 | 1 | 0 | 1 | 1 | 0 | 1 | 1 | 1 | 1 | Int | High | Int | 2 | 5 |
| P26 | 0 | 0 | 0 | 0 | 0 | 0 | 1 | 1 | 1 | Low | Low | Low-Int | 1 | 1 |
| P27 | 0 | 0 | 0 | 1 | 0 | 0 | 0 | 1 | 1 | Int | Int | Low-Int | 1 | 1 |
| P28 | 0 | 0 | 1 | 0 | 1 | 0 | 1 | 1 | 1 | Low | High | Int | 3 | 3 |
| P29 | 0 | 0 | 0 | 1 | 1 | 0 | 0 | 0 | 0 | Int | Int | Low-Int | 1 | 2 |
| P30 | 1 | 0 | 1 | 1 | 1 | 1 | 0 | 1 | 1 | Int | High | Int | 3 | 5 |
| P31 | 0 | 0 | 1 | 1 | 1 | 0 | 0 | 0 | 0 | Int | High | Low-Int | 2 | 3 |
| P32 | 0 | 0 | 1 | 1 | 1 | 1 | 0 | 0 | 1 | Int | High | High | 4 | 4 |
| P33 | 0 | 0 | 1 | 1 | 1 | 1 | 1 | 1 | 1 | Int | Int | Low-Int | 2 | 5 |

**Table 1 (continued) | (A) Baseline characteristics (B) Baseline risk stratification**

**B**

| Patient ID | IgA Isotype[b] | M-spike ≥ 3.0[c] | Immuno paresis[d] | FLC Ratio ≥ 8[e] | Evolving Disease[f] | BMPC 50–60%[g] | t(4;14) -17p +fq[h] | High-Risk cytogenetics FISH[i] | High-Risk cytogenetics FISH + Seq[j] | Mayo 2008 | IMWG 20/ 2/20 | IMWG High Precision | Number of High-Risk Criteria[k] | Total number of eligibility criteria met (+BMPC ≥ 10%)[l] |
|---|---|---|---|---|---|---|---|---|---|---|---|---|---|---|
| P34 | 0 | 0 | 0 | 1 | 0 | 0 | 1 | 1 | 1 | Int | High | Int | 2 | 2 |
| P35 | 0 | 0 | 0 | 1 | 1 | 0 | N/A | N/A | N/A | Int | Int | Low-Int | 1 | 2 |
| P36 | 0 | 0 | 1 | 1 | 0 | 0 | 1 | 1 | 1 | Int | Int | Int | 1 | 3 |
| P37 | 1 | 0 | 1 | 1 | 1 | 0 | 0 | 0 | 0 | Int | Int | Int | 1 | 4 |
| P38 | 0 | 0 | 1 | 1 | 1 | 0 | 1 | 1 | 1 | Int | High | High | 4 | 4 |
| P39 | 0 | 0 | 1 | 0 | 0 | 0 | 0 | 0 | 0 | Low | High | Low-Int | 1 | 1 |
| P40 | 0 | 0 | 1 | 1 | 1 | 0 | 0 | 1 | 1 | Int | Low | Low-Int | 2 | 3 |
| P41 | 0 | 0 | 0 | 1 | 0 | 0 | N/A | N/A | 0 | Int | Int | Low-Int | 0 | 1 |
| P42 | 1 | 0 | 1 | 1 | 1 | 0 | 0 | 1 | 1 | Int | Int | Low-Int | 2 | 4 |
| P43 | 0 | 0 | 0 | 1 | 1 | 0 | N/A | N/A | N/A | Int | Int | Low-Int | 1 | 2 |
| P44 | 0 | 0 | 1 | 1 | 1 | 0 | 0 | 0 | 1 | Int | Int | Int | 2 | 3 |
| P45 | 1 | 0 | 1 | 1 | 1 | 0 | 1 | 1 | 1 | Int | Int | Low-Int | 2 | 5 |
| P46 | 1 | 0 | 1 | 1 | 1 | 0 | 1 | 1 | 1 | Int | High | Int | 3 | 5 |
| P47 | 1 | 0 | 0 | 0 | 0 | 0 | 1 | 1 | 1 | Low | Int | Low | 1 | 2 |
| P48 | 0 | 0 | 1 | 0 | 0 | 0 | 0 | 1 | 1 | Low | Int | Low-Int | 1 | 1 |
| P49 | 0 | 0 | 1 | 1 | 1 | 0 | N/A | N/A | N/A | Int | High | Low-Int | 2 | 3 |
| P50 | 0 | 0 | 0 | 1 | 1 | 0 | 0 | 1 | 1 | Int | High | Int | 3 | 2 |
| P51 | 0 | 0 | 1 | 0 | 0 | 0 | 0 | 0 | 0 | Low | High | Low-Int | 1 | 1 |
| P52 | 1 | 0 | 1 | 1 | 0 | 0 | N/A | N/A | N/A | Int | High | Int | 1 | 3 |
| P53 | 1 | 0 | 0 | 1 | 0 | 0 | 1 | 1 | 1 | Int | Int | Low-Int | 1 | 3 |
| P54 | 1 | 0 | 0 | 1 | 1 | 0 | 0 | 1 | 1 | Int | High | Low-Int | 2 | 2 |
| P55 | 0 | 0 | 0 | 1 | 1 | 0 | N/A | N/A | 1 | Int | High | High | 4 | 2 |

FLC free light chain, BMPC bone marrow plasma cells, FISH fluorescent in-situ hybridization, Seq sequencing, Int intermediate.
[a]Cytogenetic abnormalities are compiled from FISH, scRNA-seq and WGS at screening or pre-screening timepoint. For one patient, a primary event translocation was taken from a post-screening timepoint. Cytogenetic data was unknown for six patients.
Columns[b-h] contain high-risk criteria as defined by Rajkumar et al.[5]. High-Risk cytogenetics in columns[i,j] defined as t(4;14), t(14;16), 1q gain/amp, 13q deletion/monosomy, deletion 1p, and 17p deletion.
[k]Total Number of High-Risk Criteria by Mayo 2008, IMWG 20/2/20, IMWG High Precision, Evolving Disease, and High-Risk Cytogenetics by FISH and/or Sequencing.
[l]Total Eligibility criteria met for enrollment as described in "Methods" section.

**Table 2 | Summary of adverse events reported during treatment**

| Event—no. (%) | (N = 55) | | | |
|---|---|---|---|---|
| | All grades | Grade 2[a] | Grade 3[a] | Grade 4[a] |
| Any event | 55 (100) | 20 (36) | 34 (62) | 5 (9) |
| Blood and lymphatic system disorders | | | | |
| Neutropenia | 43 (78) | 18 (33) | 14 (25) | 2 (4) |
| Thrombocytopenia | 33 (60) | 4 (7) | 4 (7) | 1 (2) |
| Leukopenia | 44 (80) | 21 (38) | 10 (18) | |
| Anemia | 33 (60) | 6 (11) | 1 (2) | |
| Lymphopenia | 15 (27) | 4 (7) | 6 (11) | 2 (4) |
| Gastrointestinal disorders | | | | |
| Diarrhea | 27 (67) | 5 (9) | | |
| Constipation | 26 (47) | 1 (2) | | |
| Nausea | 33 (60) | 3 (5) | | |
| General disorders and administration site conditions | | | | |
| Fatigue | 42 (76) | 11 (20) | 1 (2) | |
| Insomnia | 31 (56) | 3 (5) | | |
| Infections and infestations | | | | |
| Upper Respiratory Infection | 18 (33) | 15 (27) | 2 (4) | |
| Investigations | | | | |
| Alanine aminotransferase increased | 14 (25) | | | |
| Metabolism and nutrition disorders | | | | |
| Hypophosphatemia | 16 (29) | 8 (15) | 7 (13) | |
| Hypomagnesemia | 13 (24) | | | |
| Hyperglycemia | | | | 1 (2) |
| Hypokalemia | 10 (18) | | 3 (5) | 1 (2) |
| Nervous system disorders | 32 (9.0) | | | |
| Sensory peripheral neuropathy | 31 (56) | 7 (13) | | |
| Skin and subcutaneous tissue disorders | | | | |
| Maculo-papular rash | 38 (69) | 11 (20) | 5 (9) | |
| Vascular disorders | | | | |
| Peripheral edema | 31 (56) | 5 (9) | | |

[a]If applicable.

**Table 3 | Summary of responses to treatment**

| | N = 55 (%) |
|---|---|
| Best response (IMWG)—no. (%) | |
| PR or better [95% CI] | 51 (93) [92–98] |
| VGPR or better [95% CI] | 25 (45) [32–59] |
| CR[a] | 17 (31) |
| VGPR | 8 (15) |
| PR | 26 (47) |

[a]All patients with CR achieved stringent CR.

## Time-to-event analyses

The primary endpoint of PFS was defined as the development of end-organ damage per the SLiM-CRAB criteria or death. The median follow-up for all 55 patients was 50 months (range: 8–61 months). Participants who were not followed for at least 2 years yet remained alive and progression-free while on trial, were censored at the last date known progression-free. The total number of patients who remained progression-free without SLiM-CRAB criteria or death within 2 years of enrollment was 52 of 53 patients (98%) (Fig. 3A). One patient developed SLiM-CRAB progression within 2 years of enrollment after discontinuing therapy (described above). No patients developed SLiM-CRAB progression while on active therapy. The median PFS defined by SLiM-CRAB was not reached (95% CI: 57.7–NR) (Fig. 3A).

Thirty patients had biochemical progression events (Fig. 2). Biochemical progression was defined as a 25% increase in serum or urine M protein or a difference between involved and uninvolved FLC levels based on the IMWG criteria[23].

50 of 55 patients were followed for at least 2 years and remained biochemical progression-free (including the censored patients who came off therapy with or without progression) (91%; 95% CI: 80–97%, $p < 0.001$, one-sided exact binomial test). Median biochemical PFS was 48.6 months (95% CI: 39.9–NR) (Fig. 3B).

The median overall survival has not been reached (Fig. 3C). The median duration of response before developing biochemical progression was 47.4 months (95% CI: 37–NR) and the median time to biochemical progression was 49.9 months (95% CI: 39.9–NR).

## Subsequent treatments for those with biochemical progression but not SLiM-CRAB

Eleven patients started new treatments on SMM-directed clinical trials after completing protocol therapy before developing SLiM-CRAB progression. Of these, ten experienced biochemical progression while on follow-up (Fig. 2) and had an evolving pattern leading to enrollment in a subsequent clinical trial. An evolving pattern was defined as an increase of 10% in the M protein concentration or the involved free light chain concentration during the 6 months before screening for the clinical trial. Four patients met high-risk criteria per the 20/2/20 model on follow-up. These patients elected not to wait for end-organ damage on follow-up and proceeded with therapy on another trial of HR-SMM.

## Factors that impact PFS

We assessed several baseline characteristics to identify factors that define the population that benefits the most from a lenalidomide-based regimen and endpoints that can be early biomarkers for end organ damage in this population. We examined baseline IMWG 20/2/20 model risk, evolving subtype, high-risk cytogenetics, the best response to therapy, and the combination of response, high-risk cytogenetics, and MRD. These were selected based on their association with prognosis in overt MM and lenalidomide studies in SMM[24,25].

We first determined whether the baseline IMWG 20/2/20 criteria helped identify patients who would have a prolonged PFS in response to therapy. The IMWG 20/2/20 criteria stratifies patients into low, intermediate, and high-risk based on cutoffs >20, 2 g/dl, and 20% for free light chain ratio, serum M protein, and bone marrow plasma cell burden, respectively. We combined low- and intermediate-risk patients and compared them to high-risk patients. Here, we showed that the median biochemical PFS was longer but not significantly different at 57.8 months for the low-/intermediate-risk group compared to 37.2 months for high-risk patients (log-rank, $p = 0.08$) (Fig. 4A). The median duration of response before biochemical progression was significantly shorter in high-risk patients (31.6 months) than in the low/intermediate group (56.9, log-rank, $p = 0.046$) (Fig. 4B).

Evolving disease (defined in the previous section) was assessed in all participants during eligibility screening. An evolving pattern was found in 29 of 55 patients enrolled (53%; Table 1). The presence of evolving disease at baseline was similar across baseline IMWG 20/2/20 risk groups: low (56%), intermediate (48%), and high (56%) (Cochran-Armitage, $p = 0.83$). Of the patients who demonstrated an evolving pattern, only 31% ($n = 9/29$) achieved a VGPR or better, and of those who did not demonstrate an evolving pattern, 62% ($n = 16/26$) achieved a VGPR or better (Fisher's exact, $p = 0.032$).

In patients who did not demonstrate an evolving pattern, biochemical PFS was significantly longer at 53.4 months compared to

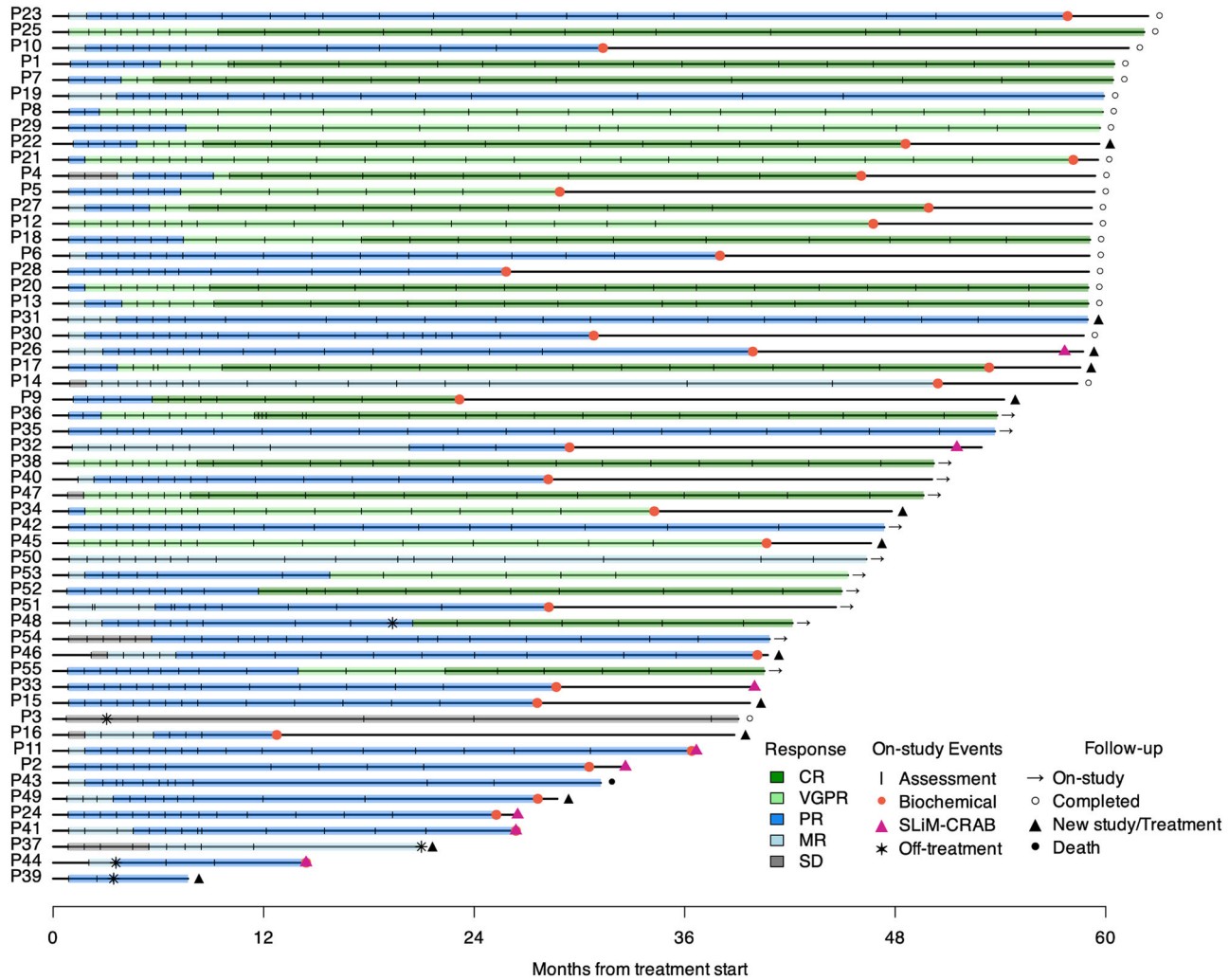

**Fig. 2 | Swimmer's plot of responses deepening over time.** Each lane represents one participant. Source data are provided as a Source Data file.

40.1 months in those with evolving disease before registration (log-rank, $p = 0.025$). Additionally, in patients who did not demonstrate an evolving pattern, the median duration of response before biochemical progression was not reached and was significantly longer than that in patients with an evolving subtype (37.9 months; log-rank, $p = 0.034$) (Fig. 4C–F).

We next examined whether patients with VGPR or CR had an improved PFS. Response is associated with PFS in overt MM[26]. However, this has not been shown in SMM, because studies using lenalidomide and dexamethasone had few cases of CR and current quadruplet regimen studies have short follow-up in SMM. We, therefore, asked whether depth of response was associated with progression in the SMM setting. Indeed, we demonstrated that patients who achieved VGPR or better had a longer median time to biochemical progression (TTP; 58.2 months) than patients who achieved inferior responses (31.3 months; log-rank, $p < 0.001$; Fig. 5A). Interestingly, this was still true in patients with high-risk cytogenetics determined by FISH only (median PFS: 31.3 months vs 53.4 months, log-rank, $p = 0.002$) or by FISH and WGS (median PFS: 30.8 months vs 53.4 months, $p < 0.001$), demonstrating that depth of response is relevant for prognostication independent of cytogenetics (Fig. 5 B, C).

All patients who achieved a CR maintained it for at least 6 months. For progression defined by SLiM-CRAB criteria which was the primary endpoint, the median PFS in the sustained CR group was not reached compared to 57 months in the non-CR groups (log-rank, $p = 0.010$) (Fig. 5D).

The median duration of response before biochemical progression in the sustained CR group was not reached at the time of analysis, while the median duration of response before biochemical progression in the non-CR group was 35.5 months (log-rank, $p = 0.0016$) (Fig. 5E). Similarly, the median biochemical PFS in the sustained CR group was not reached, compared to 38 months in the non-CR group (log-rank, $p = 0.0018$) (Fig. 5F).

**Minimal residual disease**

MRD testing was performed on 14 of 25 patients who achieved VGPR or better. At the end of therapy (EOT), 4/14 (29%) patients were MRD negative at $10^{-5}$ sensitivity. None of them developed SLiM-CRAB criteria or biochemical progression over the follow-up period. Of those who were MRD positive at $10^{-5}$ ($n = 10/14$) at EOT, six patients developed biochemical progression during the follow-up period. The 5-year PFS rate of patients who achieved MRD negativity at $10^{-5}$ was 100%, compared to 40% of MRD-positive patients (log-rank, $p = 0.051$) (Fig. 5G). MRD testing at both C9 and EOT was available in a subgroup of 8 individuals (Supplementary Fig. S2). Of these, 4/8 patients were MRD negative at $10^{-5}$ at C9, but only 1 patient remained MRD negative at EOT. The 3 patients that converted from C9 MRD negative at a sensitivity of $10^{-5}$ to positive at EOT developed biochemical progression during the follow-up period. Four patients who were MRD positive

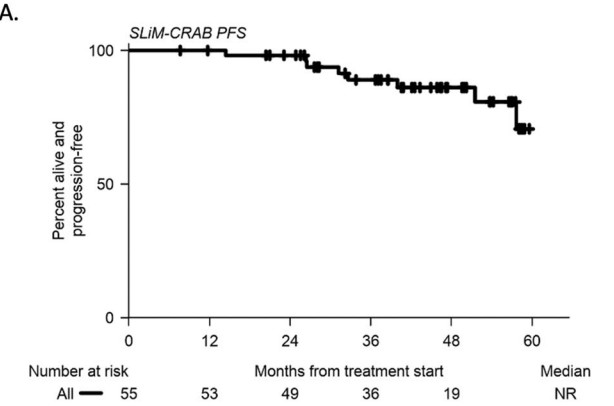

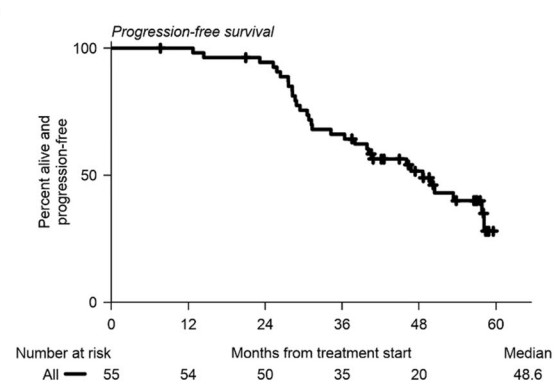

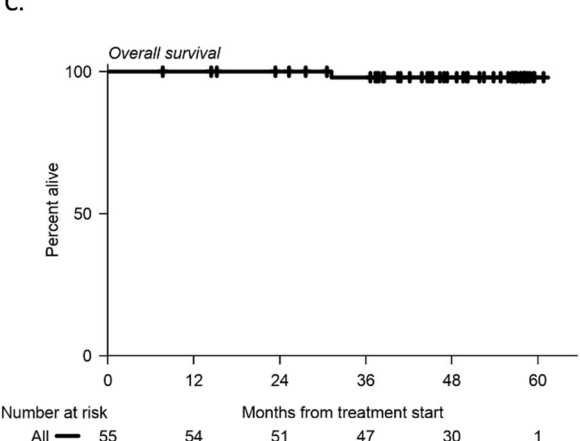

**Fig. 3 | Progression-free and overall survival.** Progression defined by **A** SLiM-CRAB criteria **B** Biochemical progression **C** Overall survival. NR not reached. Source data are provided as a Source Data file.

at $10^{-5}$ at C9 remained so at EOT. Further, MALDI-TOF mass-spectrometry showed potential as a more sensitive method for detecting residual disease compared to serum immunofixation, with negative results at the end of treatment (EOT) potentially associating with longer PFS (see Supplementary Data, Supplementary Figs. S3, S4, Supplementary Table S2).

## Single-cell RNA sequencing reveals tumor-intrinsic and extrinsic factors associated with disease progression

To identify biological predictors of response and resistance to IxaRD in baseline samples from patients with HR-SMM, we performed scRNA-seq of CD138+ and CD138- cells and WGS on 22 patients enrolled on the study (14 biochemical progressors, 8 non-progressors) and 11 healthy donors (Fig. 6A). Among these progressors, 5 of 14 met SLiM-CRAB criteria (Fig. 6A).

We identified 125,572 total plasma cells (Patients: malignant, $n = 50,591$; normal, $n = 1876$; HD: $n = 73,105$) (Fig. 6B–D). To identify tumor-intrinsic transcriptomic alterations associated with biochemical progression post-IxaRD in patients with HRSMM, we performed differential expression analysis comparing tumor cells from patients who progressed ($n = 10$) to those who did not ($n = 3$). To restrict the influence of individual tumors on the analysis, we randomly sampled the same number of tumor cells from progressors and non-progressors (~2100 per group) and balanced the contribution of each individual to the group (progressors: ~210 cells

per patient; non-progressors: 700 cells per patient; Fig. 6E). We observed consistently higher expression levels of MHC class I genes (*HLA-A, HLA-B, HLA-F, HLA-E, B2M*) in tumor cells from non-progressors to progressors (Fig. 6E). To validate this observation, we analyzed bulk RNA-seq data from an external cohort of MM patients treated with a Bortezomib-based regimen, a different proteasome inhibitor (PADIMAC, GSE116234). This analysis confirmed that tumor cells from patients who respond well to proteasome inhibition show higher levels of MHC-I genes (Fig. 6F). Notably, the PADIMAC study did not use Lenalidomide in combination with proteasome inhibition, which was the case in our study. This supports the notion that the observed association with MHC-I levels is related to proteasome inhibition specifically rather than a particular combination.

These results suggest that antigen presentation via MHC-I by tumor cells may be important for response to proteasome inhibition and implicate the immune system in mediating response. While the immune system has been thought to potentiate the effects of proteasome inhibition through increased neoantigen presentation by immune cells following tumor cell killing, it has not been previously linked to tumor-intrinsic transcriptomic alterations[27]. To pursue this hypothesis, we turned our attention to the cytotoxic CD8+ T cell compartment. We analyzed 51,326 T cells, including 5045 GZMK+ and 9026 GZMB+ cytotoxic T cells (Fig. 6G). We observed that the GZMK+ compartment was significantly more clonally expanded in

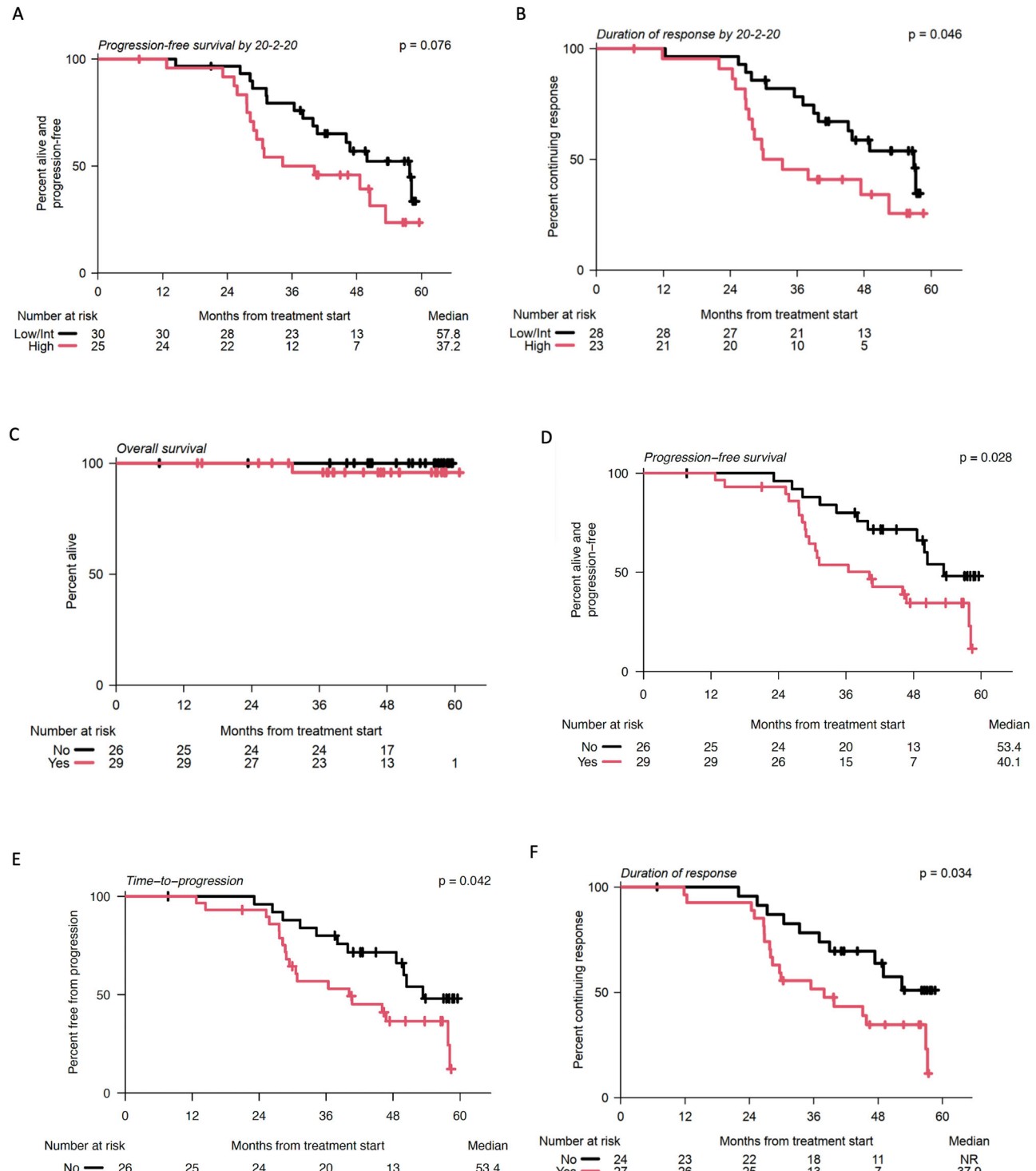

**Fig. 4 | Progression-free survival. A** Kaplan-Meier curve of response duration stratified by 20/2/20 risk groups. **B** Kaplan-Meier curve of progression stratified by 20/2/20 risk groups. Survival distributions were compared using two-sided log-rank tests. *Outcomes stratified by the presence of evolving disease prior to treatment.*

**C** Overall survival **D** Biochemical progression-free survival **E** Time-to-progression and **F** Duration of response. NR not reached. Survival distributions were compared using two-sided log-rank tests. Source data are provided as a Source Data file.

progressors compared to non-progressors (bootstrapping, two-sided $p < 0.001$) (Fig. 6H), but this was not seen in the GZMB+ compartment. By comparing the frequency of the GZMB+ phenotype within clonally expanded T cells from progressors and non-progressors, we observed a significant decrease in differentiation towards the GZMB+ phenotype in progressors (Wilcoxon, two-sided $p = 0.048$) (Fig. 6I, J). This suggests that clonally expanded effector

cytotoxic T cells from progressors may be less mature compared to non-progressors. To gain more insight into the functionality of clonally expanded GZMB+ cytotoxic T cells, we performed differential expression analysis between progressors and non-progressors. Non-progressors had higher expression of cytotoxicity and activation genes (*TYROBP, CXCR4, AP-1*) in clonally expanded GZMB + T cells compared to progressors (Fig. 6K). While progressors

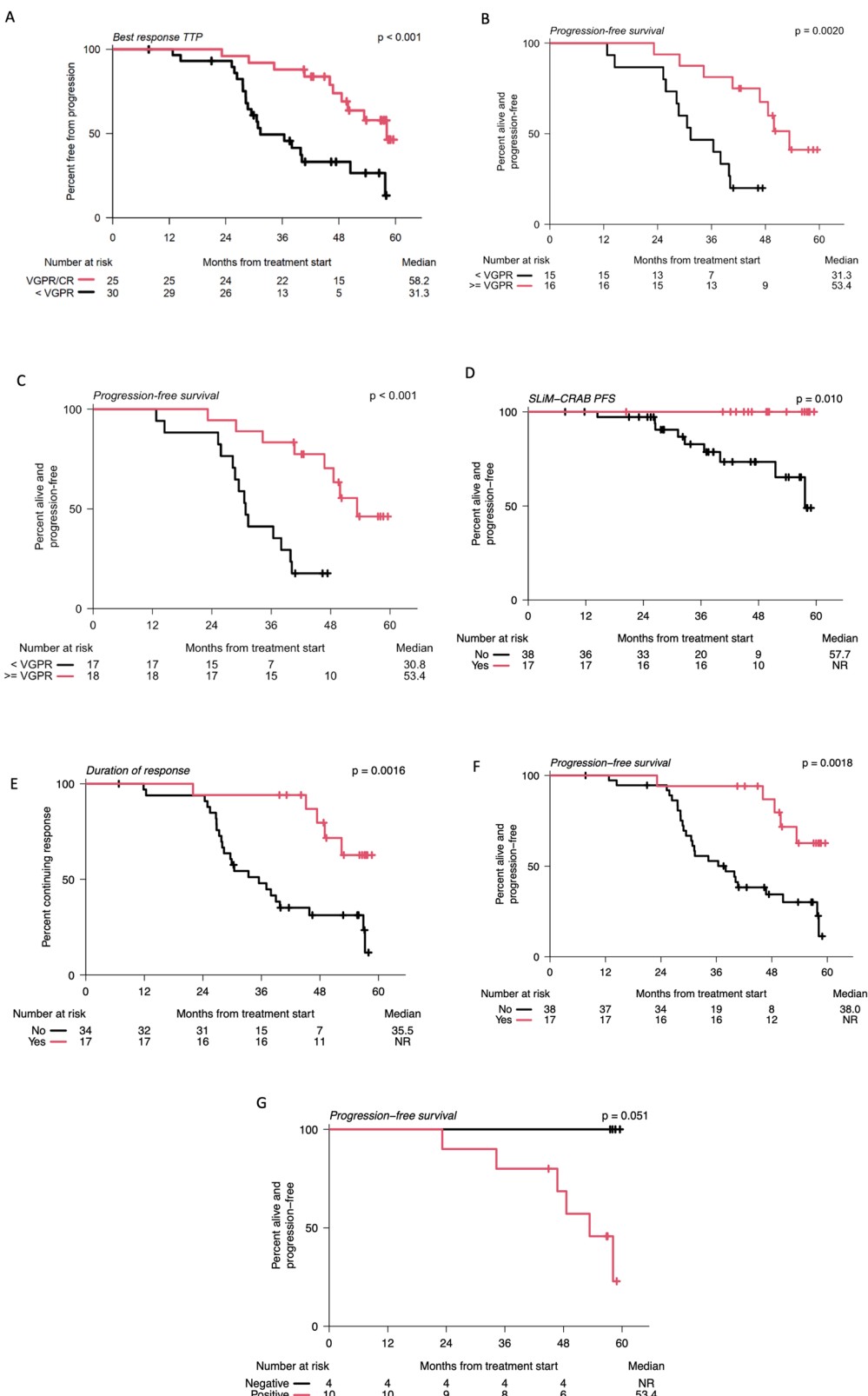

**Fig. 5 | Kaplan Meier curve of progression-free survival.** Based on (**A**) Time to progression based on depth of response of VGPR of greater. $p = 8.0e\text{-}4$ VGPR = Very good partial response (**B**) PFS by depth of response of VGPR of greater amongst cytogenetically high-risk patients (high-risk cytogenetics identified by FISH alone). **C** PFS by depth of response of VGPR of greater amongst cytogenetically high-risk patients (High-risk patients identified by FISH, WGS, or single-cell RNA sequencing.) $p = 3.1e\text{-}4$. **D** Kaplan-Meier curve of progression-free-survival stratified by patients

who achieved sustained CR with progression defined by SLiM-CRAB criteria. CR complete response. **E** Kaplan-Meier curve of duration of response stratified by patients who achieved sustained CR. **F** Kaplan-Meier curve of Biochemical progression-free-survival stratified by sustained CR status. **G** Kaplan Meier curve of progression-free survival based on MRD status. NR not reached. Survival distributions were compared using two-sided log-rank tests. Source data are provided as a Source Data file.

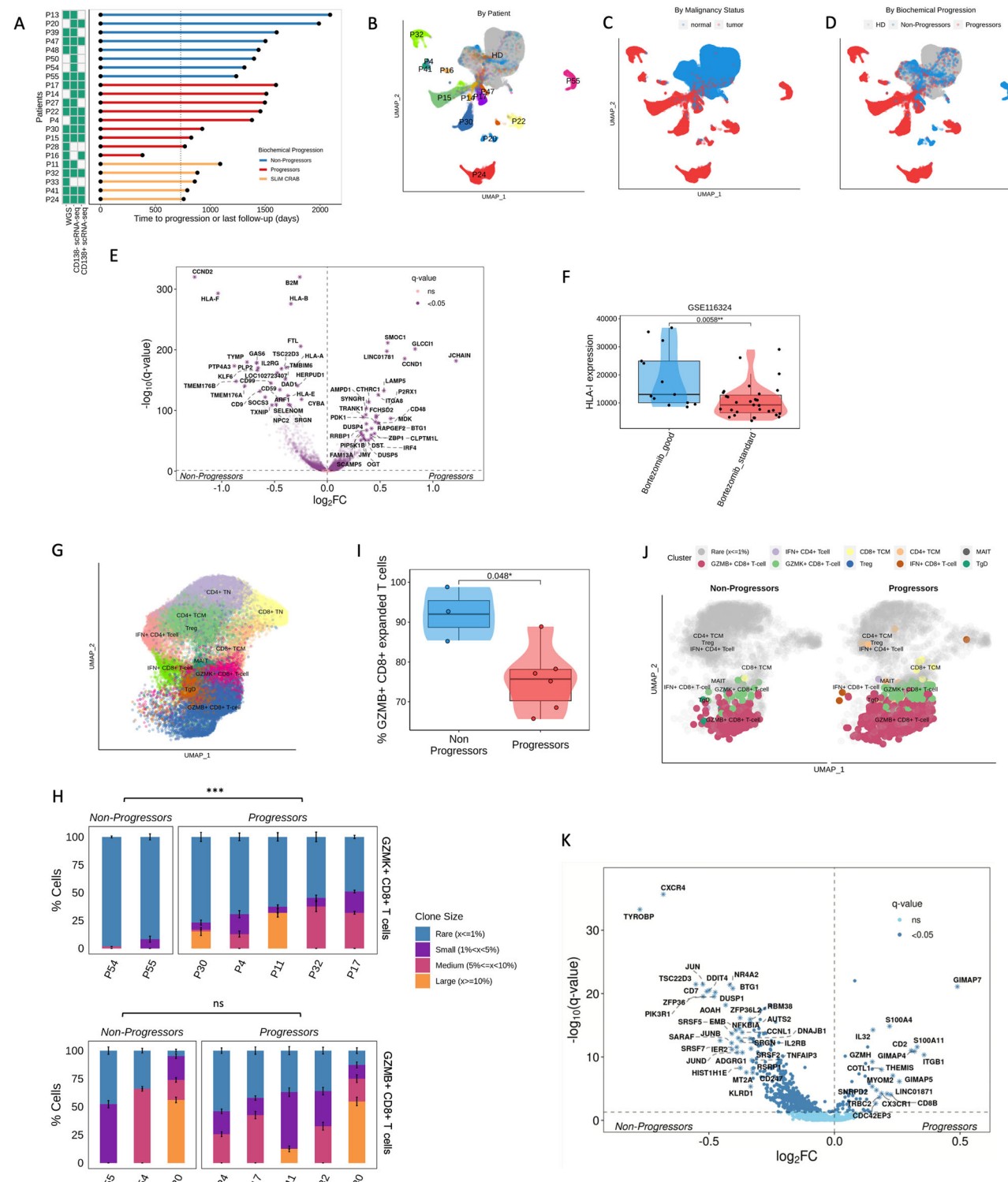

showed increased *GIMAP* genes (apoptosis) and *THEMIS* (suppressed response), they also had higher maturation markers (*ITGB1, CX3CR1*), suggesting that the functionality of clonally expanded GZMB+ cytotoxic T cells may be more nuanced.

Overall, this suggests that clonally expanded cytotoxic T cells from progressors are phenotypically less mature, and potentially less functional, compared than those from non-progressors, which could be related to the observed differences in MHC-I expression in tumor cells.

## Discussion

Optimal management of HR-SMM continues to evolve with novel approaches under investigation to deliver highly effective therapy to prevent end-organ damage and limit toxicity. IxaRD demonstrated high response rates and many patients experienced deep responses, with VGPR or better in nearly half of the patients treated. No patients developed MM during study therapy and the all-oral regimen was well-tolerated overall. The response rates and safety were comparable to that of IxaRD in newly diagnosed and relapsed MM[28,29], however, in this

**Fig. 6 | Single-cell RNA sequencing results. A** Time to progression or last follow-up for patients with sequencing data (one patient per row, grouped by progression status). Green boxes indicate baseline sequencing data available for each patient. Vertical dotted line denotes end of treatment. **B** Uniform manifold approximation and projection (UMAP) of patient and HD plasma cells that passed quality filtering. Colors indicate individual patients or HD as a group **C** UMAP colored by malignancy status (tumor vs normal cells). **D** UMAP colored by biochemical progression status. **E** Volcano plot of differential gene expression in tumor cells of progressors (*n* = 10) and non-progressors (*n* = 3). Two-sided *p* values were computed with Wilcoxon rank-sum test and corrected using the Benjamini-Hochberg approach. Stars = top 30 genes on either side of the volcano with *q* < 0.05. **F** Mean HLA class I gene expression from tumor bulk RNA-seq from the PADIMAC cohort who responded (Bortezomib_good, *n* = 13) or did not respond (Bortezomib_standard, *n* = 31) to Bortezomib treatment. Box: 1st quartile, median, 3rd quartile; whiskers: ±1.5*interquartile range. The *p* value was computed with two-sided Wilcoxon rank-sum test. **G** UMAP of T cell subtypes (**H**) Average proportion of clonotypes belonging to four clone size categories (rare, <1%; small, ≥1% and <5%; medium, ≥5% and <10%; large: ≥10%) per patient in baseline BM CD138- samples. Error bars = SD from 100 iterations. *P* values were obtained using two-sided bootstrapping test to compare the mean proportion of rare clones between progressors and non-progressors. In all types of statistical analysis values of *p* < 0.05 were considered significant (**p* < 0.05; ***p* < 0.01; ****p* < 0.001; *****p* < 0.0001). **I** GZMB+ CD8+ T effector memory cell abundance in expanded clones from non-progressors and progressors. Plot statistics same as (**F**). **J** UMAP of baseline BM T cells from progressors and non-progressors with matched TCR data. T cells with expanded clones (frequency > 1%) are colored by cell type. Gray = cells with rare clonotypes **K** Volcano plot of differential gene expression of clonally expanded CD8+ TEM in progressors and non-progressors. Statistics same as (**E**). HD healthy donors, P patient, WGS whole-genome sequencing, HRD hyperdiploidy. Source data are provided as a Source Data file.

study median biochemical and overall PFS (48.9 and NR months, respectively) were both significantly longer than in newly diagnosed MM (median PFS: 35.3 months)[28]. SMM therapy was also limited to 2 years with no maintenance therapy. Because of the long follow-up (median: 50 months) and the depth of response achieved in half of the patients, we had an opportunity to investigate critical questions about early intervention in HR-SMM.

The importance of depth of response in HR-SMM trials is an open question. Previous studies using lenalidomide alone or lenalidomide and dexamethasone demonstrated a long-term benefit and delayed end organ damage despite PR being the best response[8,24]. If immune equilibrium may be re-instated in those patients, deep remission is not required. However, studies with "curative intent" argued that achieving a depth of response similar to overt MM is the optimal way to achieve long-term remission[14]. Here, we show for the first time that a depth of response of sustained CR and VGPR or greater correlated with TTP for both SLiM-CRAB and biochemical progression, suggesting that deep response has similar impacts in SMM and MM. This was highlighted in patients with cytogenetically high-risk disease, where PFS was only 30.8 months in patients with less than VGPR.

We asked whether MRD is associated with improved PFS. Although the numbers were small, we were able to see near-significant differences between the PFS of MRD-negative cases and those who did not achieve or sustain MRD-negativity. Ongoing studies with deeper remissions with quadruplet therapy or immunotherapy will help address this question more definitively.

We also examined the correlation of biochemical progression with SLiM-CRAB criteria. Thirty total patients developed biochemical progression (27 after completion of therapy), of which, 8 also developed SLiM-CRAB with overt MM. The strong correlation between biochemical progression and the development of SLiM-CRAB indicates that biochemical progression can be used as a surrogate for end organ damage in HR-SMM. This observation needs further confirmation to be used as a primary endpoint in clinical trials of HR-SMM.

Eleven patients started new treatments on SMM-directed trials after completing protocol therapy before developing SLiM-CRAB progression. Of these, ten experienced biochemical progression while on follow-up and had an evolving pattern leading to enrollment in a subsequent clinical trial. Four of these patients met 20/2/20 high-risk criteria, consistent with our observation that patients who opt to be treated for HR-SMM do not want to wait for the development of end-organ damage before re-initiating therapy, and conversations considering biochemical progression as an end-point of therapy rather than SLiM-CRAB[13]. This brings up a critical question of whether patients receiving re-treatment on an SMM trial should be assigned to the same cohort as participants who were never exposed to therapy or a separate cohort. Additional guidance is necessary to determine which patients would be suitable for further (relapsed) SMM-directed therapy vs. MM induction therapy.

Interestingly, the use of 20/2/20 criteria in our study demonstrated that responses were similar across all risk criteria, but the PFS was shortest for those who were high-risk per 20/2/20. This indicates that high-risk 20/2/20 patients may resemble MM and may require more aggressive therapy such as quadruplet, immunotherapy, or maintenance therapy beyond a 2-year time-limited therapy.

Genomic alterations are critical to better define patients who are truly at risk of progression and may benefit from more aggressive therapy. Here, WGS and scRNAseq helped identify participants with high-risk cytogenetics who were not identified by FISH alone. We did not include MAPK and DNA repair pathway mutations in analyzing the impact on outcomes because we did not have data available on all patients. With additional data, we were able to define a group with high-risk cytogenetics and poor response to therapy who showed the worst outcome.

We performed an integrative study combining WGS and scRNA-seq of tumor and immune cells. We observed significantly higher MHC class I gene expression (GEX) in tumor cells from non-progressors than progressors. Progressors had a reduced proportion of the GZMB+ phenotype within their clonally expanded T cells, suggesting their clonally expanded T cells have a less mature cytotoxic profile. Our findings suggest that immune dysfunction related to changes in tumor intrinsic antigen presentation may contribute to suboptimal responses in proteasome inhibitor therapy.

This is the first trial studying an all-oral triplet combination of IxaRD in HR-SMM, demonstrating substantial efficacy and identifying several factors critical in the outcome of patients: depth of response, evolving pattern, MRD negativity, and high risk by 20/2/20. Future studies using quadruplet therapy or immunotherapy may alter these conclusions.

## Methods
The research study was approved by the Dana-Farber/Harvard Cancer Center institutional review board (protocol number DFCI 16-313) and complied with all relevant ethical and legal regulations. All participants gave written informed consent.

### Eligibility criteria
Patients enrolled on the I-PRISM study met eligibility for HR-SMM based on criteria by Rajkumar[22] as follows: bone marrow plasmacytosis ≥10% and ≤60% and any one or more of the following: serum M protein ≥3 g/dL, IgA isotype, immune paresis with reduction of 2 uninvolved immunoglobulins, serum involved/uninvolved ratio of ≥8 but ≤100, bone marrow plasmacytosis 50–60%, abnormal plasma cell immunophenotype, high-risk FISH defined as t(4;14), del 17p or 1q gain, increased circulating plasma cells, MRI with diffuse abnormalities or 1 focal lesion, PET/CT with one focal lesion with increased uptake without underlying osteolytic bone destruction and urine monoclonal light chain excretion ≥500 mg/24 h. Patients with evidence of

SLiM-CRAB criteria were excluded (bone marrow plasmacytosis > 60%, light chain ratio > 100)[30]. Patients were required to have an ECOG performance status of ≤2 and adequate hematologic and organ function prior to enrollment. A creatinine clearance of ≥30 mL/min was required.

## Treatment cegimen

Patients were treated on an outpatient basis with 9 cycles of induction therapy consisting of ixazomib 4 mg given orally on days 1, 8, and 15, in combination with lenalidomide 25 mg administered orally on days 1 through 21 and dexamethasone 40 mg administered orally on days 1, 8, 15, and 22 of a 28-day cycle. The induction phase was followed by maintenance therapy, consisting of ixazomib 4 mg on days 1, 8, and 15 and lenalidomide 15 mg on days 1–21 without dexamethasone for another 15 cycles. Treatment duration was a total of 24 cycles (2 years).

## Objectives and end points

The primary objective was to determine the proportion of patients with HR-SMM who are progression-free 2 years after receiving trial therapy. Progression for the primary endpoint was defined as the development of end organ damage per SLiM-CRAB criteria. Secondary endpoints included response rate, duration of response, PFS, safety of the combination therapy, and MRD-negativity. Mass spectrometry analysis was also performed to correlate with IMWG response and MRD status.

## Minimal residual disease assessment and mass spectrometry

MRD was assessed by the next-generation sequencing (NGS) clonoSEQ assay (Adaptive Biotechnologies, Seattle, WA, USA) on bone marrow samples from patients who achieved at least a very good partial response (VGPR) at cycle 9 (C9) or later. When BM was available, a second MRD assessment was performed to confirm the MRD result. Quality control (QC) of NGS results was performed as previously described[31].

Residual disease after therapy was also assessed by matrix-assisted laser desorption ionization-time of flight (MALDI-TOF) mass spectrometry to quantify M proteins (Binding Site Group, Birmingham, UK), and Optilite free light-chain assay to quantify IgG, IgA, IgM, and serum free light chains. EXENT-iQ software (Binding Site Group, Birmingham, UK) was used for quantitative analysis of detected M proteins. The lower limit of accurate M protein quantification by MALDI-TOF mass spectrometry was 0·015 g/L. Serum (500 µL) was tested using the EXENT system and immunoglobulin (Ig) isotype assay[32]. The unique mass/charge (m/z) (±5) of a monoclonal light chain and the isotype were defined at baseline and tracked in each patient's samples after therapy at the end of induction (C9) and when available during maintenance (2–3 months after C9) and at EOT. MALDI-TOF was performed at baseline before treatment in 48 patients out of 55.

## Evolving disease pattern

Evolving disease was assessed in all participants as part of the eligibility criteria and was defined as a 10% or greater increase in either the M-spike concentration or the involved free light chain concentration within the 6 months preceding clinical trial screening. The median time of the lab result was 107 days prior to the screening date with an interquartile range of 120 days.

## DNA isolation and library construction

Genomic DNA isolation was carried out using the Monarch Genomic DNA Purification Kit (New England Biolabs), with tumor (PCs) and normal (PBMCs) yields quantified by Qubit 3.0 fluorometer (Thermo Fisher Scientific). 50–100 ng was taken forward for DNA sequencing library preparation using the NEBNext Ultra II FS DNA Library Prep kit (New England Biolabs) with unique dual index adapters (NEBNext Multiplex Oligos) according to manufacturers' instructions. Final library fragment sizes were assessed using the BioAnalyzer 2100

(Agilent Technologies), and yields were quantified by Qubit 3.0 fluorometer (Thermo Fisher Scientific) and qPCR (KAPA Library Quantification Kit).

WGS and genomic data analysis.

WGS was available for 17 patients. Final sample libraries were normalized and pooled before WGS was performed on Illumina NovaSeq 6000 S4 flow cells, 300 cycles 2x150bp paired-end reads, at the Genomics Platform of the Broad Institute of MIT and Harvard. WGS analysis was performed on an in-house cloud-based HPC system for copy-number and structural variant analyses, as previously described[33]. Sequencing reads were aligned to the GRCh38 reference genome with the bwa mem v0.7.7 algorithm[34], duplicate reads were marked with MarkDuplicates from picard v1.457, indels were realigned with GATK 3.4 IndelRealigner, and base qualities were recalibrated with GATKBQSR. WGS analysis was performed with the Cancer Genome Analysis workflow from the Cancer Program at Broad Institute of MIT and Harvard on an in-house cloud-based HPC system. Small indels were called with Strelka2 and were filtered (i) against a panel of normals (PoN), (ii) for potential technical artifacts (oxoG), and (iii) for multiple alignments with BLAT[35]. After copy-number normalization with AllelicCapSeg, ABSOLUTE solutions were manually reviewed to estimate mutation CCF, purity, and ploidy of tumor samples[36]. Structural variants were detected and filtered as previously described[37].

## Patient sample processing for scRNA-seq

Whole bone marrow aspirates (5–20 mLs) were drawn into EDTA preservative tubes and kept at 4 °C before being processed within 6 h of collection. Bone marrow mononuclear cells were isolated from aspirates using density gradient centrifugation. Briefly, the bone marrow was filtered to remove any clots or bone debris, diluted 1:3 with 1 × Phosphate-Buffered Saline (PBS), and gently poured above 15mLs of Ficoll-Paque density gradient medium within a SepMate™ PBMC isolation tube (StemCell Technologies, Cat # 85450). After centrifugation at 1200 rcf/g for 15 min, PBMCs were poured away from unwanted cells and washed twice with ice-cold 1 × PBS prior to plasma cell enrichment. Plasma cells were enriched from the bone marrow samples using CD138 magnetic bead separation using single-column positive selection (Miltenyi Biotec, Cat #130-097-614).

## Single-cell RNA/V(D)J library construction and sequencing

Sequencing libraries were prepared from BM samples, including 5′ scRNA-seq libraries and scBCR-seq and scTCR-seq libraries. After MACS enrichment, all CD138+ and CD138- cells were centrifuged at 300 rcf/g for 5 min and washed twice with an ice-cold 0.1% Ultrapure Bovine Serum Albumin (BSA)/PBS wash buffer. Subsequently, cells were either subjected to volume reduction or diluted based on cell counts obtained using both a hemocytometer and a Countess automated cell counter (ThermoFisher), and cell mixtures were adjusted to obtain optimal cell densities for achieving maximal cell recovery. Samples, Gel bead-in-EMulsion (GEMs), and partitioning oil were loaded into a Next GEM Chip K microfluidic device and placed in a Chromium Controller instrument for downstream single-cell encapsulation and recovery. All GEM generation/barcoding, post GEM RT clean-up/cDNA amplification, and 5′ GEX library construction steps were completed using the Chromium Next GEM Single Cell 5′ Reagent Kit v2 (Dual Index) and Library Construction Kit, according to the manufacturer's instructions. cDNA generated was also subsequently subjected to V(D)J amplification using Chromium Single Cell Human BCR Amplification Kit and Library Construction Kit for scBCRseq and Chromium Single Cell Human TCR Amplification Kit and Library Construction Kit for scTCR-seq, according to the manufacturer's instructions. The quality of sample libraries was assessed based on library trace fragment sizes and patterns at numerous points throughout the protocol including cDNA generation, BCR V(D)J pre-amplified cDNA generation, TCR V(D)J pre-amplified cDNA generation,

final GEX, final BCR and final TCR construction steps using a High-Sensitivity DNA Analysis Kit and the Bioanalyzer 2100 instrument (Agilent Technologies). Final GEX, BCR, and TCR library quantification were performed using Quant-iT Picogreen dsDNA Assay Kit (Invitrogen) before preparing single pools for sequencing. Pooled libraries were sequenced on NovaSeq S4 flow cells at the Genomics Platform of the Broad Institute of MIT and Harvard (Cambridge, MA).

### Single-cell RNA/V(D)J data processing
CD138+ scRNA-seq and paired single-cell BCR sequencing was available for 13 patients. CD138- scRNA-seq was available for 19 patients and 9 of these patients had single-cell TCR sequencing. 12 patients had paired CD138+ and CD138- scRNA-seq. Paired WGS and CD138 + RNA sequencing was available for 10 patients. In all 20 patients had either CD138+ or CD138- scRNA-seq sequencing available.

### scRNA-seq analysis
CellRanger mkfastq (v5.0.1) was used to generate FASTQ files[38]. GEX matrices were generated by CellRanger count (v6.0.1) with the genome reference (refdata-gex-GRCh38-2020-A; 10X Genomics)[38]. To remove ambient RNA, CellBender (v0.2.0) was run on the GEX matrices with the target false positive rate cutoff of 0.01. Poor quality cells with >15% mitochondrial GEX, either <200 detected genes, >5000 detected genes, <400 UMIs, or >50,000 UMIs were filtered out. Three doublet tools, Scrublet (v0.2.3), scDblFinder (v1.8.0), and scds (1.10.0) were used to calculate multiplet scores[39–41]. CellRanger vdj (v6.0.1) was used on FASTQ files with the VDJ reference file (refdata-cellranger-vdj-GRCh38-alts-ensembl-5.0.0)[38]. Downstream analysis was performed using the Seurat 4.1[42]. For immune cells (CD138 neg), integration was performed using Harmony (v0.1.0) and correcting for sample ID[43]. Immune cells were annotated based on a list of established expression markers, as well as cluster-specific markers obtained through differential expression analysis, as previously described[16].

### Malignant plasma cell identification
Plasma cells were first identified based on expressing key lineage markers, such as SDC1 (encoding CD138), CD38, XBP1, PRDM1, IRF4, and TNFRSF17 (encoding BCMA). Cell barcodes that were determined to correspond to plasma cells were considered for downstream analysis. Tumor cells were identified based on belonging to the largest expanded BCR clone and clustering separately from normal plasma cells. For all clones, the isotype of the malignant clone matched that detected clinically via IFE.

We annotated 50,591 patient cells as tumor and the rest (n = 1876) as healthy plasma cells based on our tumor cell annotation protocol. The median number of tumor cells per patient was 2148.

### Single-cell TCR sequencing, data processing, and analysis
We performed single-cell TCR sequencing on CD138- samples with available scRNA-seq data (n = 9). Complementary DNA generated from barcoded CD138- immune cells using Chromium Single Cell 5' GEX and V(D)J enrichment kits by 10X Genomics was subjected to V(D)J enrichment and library preparation and sequenced on a NovaSeq 6000 instrument at the Genomics Platform of the Broad Institute of MIT & Harvard.

CellRanger vdj v6.0.1 was used to extract FASTQ files and produce clonotype matrices[38]. When multiple alignments were called for a single chain, the alignment with the most UMIs was selected, and when multiple chains were called for a single cell barcode, the top two chains in terms of UMI counts were selected.

### TCR clone size assignment
To categorize TCR clones based on their size or abundance within a sample, we employed a downsampling approach to account for potential sampling bias. We randomly sampled 100 T cells from each

patient sample 100 times. Within each iteration, we calculated the proportion of each unique TCR clone compared to the total T cells sampled.

Finally, for each clone, we averaged its proportional abundance across all iterations to obtain a more stable estimate of its size category. Categories were defined as: Rare: ≤1%, Small: >1% and <5%, Medium: ≥5% and <10%, Large: ≥10%.

Clone-size-category proportion estimation within cell subtypes per patient sample:

To estimate the overall proportion of clones belonging to each size category within a cell subtype per patient sample, we again used downsampling. We randomly sampled 100 T cells from each patient cell subtype sample 100 times. In each iteration, we counted the frequency of each clone size category within the downsampled set. Finally, we averaged the frequency of each category across all iterations and these averaged frequencies were then renormalized before plotting. Two-sided p values were computed by bootstrapping with 1000 iterations.

### Copy number analysis from scRNA sequencing
Copy number abnormalities on tumor cells were inferred using Numbat with default parameters (v1.1.0)[44]. Allelic data was collected from plasma cells and B cells and a panel of 1200 plasma cells from 11 healthy donors was used as expression reference.

### Differential expression analysis
Droplets with <20,000 reads per cell were discarded, and 20,000 reads per cell were downsampled across the entire CD138pos scRNA-seq cohort. To control the influence of individual tumors on the analysis, as well as control for the lower number of non-progressors in this sub-cohort (n = 3, compared to 10 progressors), we randomly sampled the same number of tumor cells from progressors and non-progressors (~2100 per group) and balanced the contribution of each individual towards the group (progressors: 210 cells per patient as one progressor patient had fewer than 210 cells (n = 72) after downsampling; non-progressors: 700 cells per patient). We repeated this process leaving out the one progressor patient with a low number of cells this time considering only 9 progressor patients and 3 non-progressors and once again balancing the contribution of each individual towards the group (progressors n = 225, non-progressors n = 675 per patient). No difference in results was observed.

### Statistical analysis
The primary endpoint was evaluated using an exact binomial test and reported as a percent with 95% exact binomial confidence intervals. The study was a single-stage, binomial design with 50 vs 70% (null vs alternative) progression-free rate targeting 5% and 10% type-I and type-II errors, respectively. The critical value to reject the null hypothesis of 50% was 33 of 53 (62.26%). Categorical secondary endpoints such as response and progression were reported as percents with 95% exact binomial confidence intervals. The Cochran-Armitage test for trend was used to evaluate responders vs non-responders for ordered 20/2/20 risk groups. Time-to-event endpoints including progression-free and overall survival were estimated using the Kaplan-Meier (KM) method and reported as medians with 95% confidence intervals. Survival distributions were compared using log-rank tests. Median follow-up time was estimated using the reverse KM method.

Correlative endpoints, including GEX, were compared between progressors and non-progressors using Wilcoxon rank-sum tests; p values were corrected using the Benjamini-Hochberg method for multiple hypothesis testing. Concordance between NGS MRD and baseline M protein results were assessed using Cohen's kappa statistic and Pearsons's product moment correlation coefficient, respectively. Data are represented as the mean ± standard error of the mean (s.e.m.) or standard deviation (s.d.) as indicated in the figure legends. In all

types of statistical analysis values of $p < 0.05$ were considered significant (*$p < 0.05$; **$p < 0.01$; ***$p < 0.001$; ****$p < 0.0001$).

Statistical analyses and scRNAseq analyses were performed using R version 4.1. Additional packages were used for survival analyses (survival, v3.5)[45,46] and agreement statistics (irr, v0.84)[47].

## Reporting summary
Further information on research design is available in the Nature Portfolio Reporting Summary linked to this article.

## Data availability
Raw single-cell RNA and TCR sequencing data generated for this study have been deposited in the dbGaP under accession code: https://www.ncbi.nlm.nih.gov/projects/gapprev/gap/cgi-bin/study.cgi?study_id=phs003827.v1 The processed gene expression matrices along with VDJ-TCR and VDJ-BCR data are available at Zenodo, Deeper Response Predicts Better Outcomes in High-Risk-Smoldering-Myeloma using Ixazomib-Lenalidomide-Dexamethasone Therapy: Phase II Clinical Trial (zenodo.org). The source data generated in this study are provided in the Supplementary Information/Source Data file. Source data are provided with this paper.

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

## Acknowledgements

The authors thank the patients and their families for participating in this study. We also thank Dr. Yoshinobu Konishi for helpful discussions. Anna V. Justis, Ph.D., a medical writer employed by Dana-Farber Cancer Institute, edited this manuscript. Takeda and Celgene (Bristol Myers Squibb) provided support for the clinical trial. These funders reviewed the final manuscript and approved its publication; they were not involved in the conceptualization, design, data collection, analysis, or preparation of the manuscript. Funding was also provided by the Dr. Miriam and Sheldon G. Adelson Medical Research Foundation, Cancer Research UK, and the NIH (R35CA263817 awarded to I.M.G.).

## Author contributions

O.N.: Writing—Original Draft, Writing—Review & Editing. M.P.A.: Writing—Original Draft, Investigation, Formal analysis, Visualization, Writing—Review & Editing. R.A.R.: Formal analysis, Visualization, Writing—Review & Editing. M.T: Formal analysis, Visualization, Writing—Review & Editing. S.M.: Writing—Review & Editing. J.A. Writing—Review & Editing, Supervision. L.B.: Investigation, Writing—Review & Editing. A.D. Writing—Review & Editing, Supervision. H.E.: Investigation, Writing—Review & Editing. M.B.: Investigation, Writing—Review & Editing. E.D.L.: Investigation, Writing—Review & Editing. J.P.L.: Writing—Review & Editing. G.B.: Writing—Review & Editing. E.O.: Writing—Review & Editing. T.W.: Writing—Review & Editing. J.T.: Writing—Review & Editing. K.A.: Writing—Review & Editing. P.G.R.: Writing—Review & Editing. G.G.: Writing—Review & Editing. L.T.: Writing—Review & Editing, Supervision. R.S.P.: Writing—Review & Editing, Supervision. I.M.G.: Conceptualization, Resources, Writing—Review & Editing, Supervision, Funding Acquisition.

## Competing interests

O.N. reports research support from Takeda and Janssen; advisory board participation for Bristol Myers Squibb, Janssen, Sanofi, Takeda, and GPCR therapeutics; and honorarium from Pfizer. M.B. reports consultancy for Janssen, BMS, Takeda, Epizyme, Karyopharm, Menarini Biosystems, and Adaptive. G.B. reports consultancy for Prothena. E.O. reports advisory board participation and/or honoraria from Janssen, BMS, Sanofi, Pfizer, Exact; consulting for Takeda; and steering committee participation with Natera. K.A. reports consultancy for AstraZeneca, Janssen, and Pfizer; and board participation and stock options for Dynamic Cell Therapies, C4 Therapeutics, Next RNA, Oncopep, Starton, and Window. P.G.R. reports advisory board participation or consulting for Celgene (BMS), GSK, Karyopharm, Oncopeptides, Regeneron, Sanofi, and Takeda; and research grants from Oncopeptides and Karyopharm. R.S.P. is a co-founder, equity holder, and consultant for PreDICTA Bioscience. I.M.G.: Consulting/Advisory role: AbbVie, Adaptive, Amgen, Aptitude Health, Bristol Myers Squibb, GlaxoSmithKline, Huron Consulting, Janssen, Menarini Silicon Biosystems, Oncopeptides, Pfizer, Sanofi, Sognef, Takeda, Binding Site a part of Thermo Fischer Scientific, Window Therapeutics, and 10X Genomics; speaker fees from Vor Biopharma and Veeva Systems, Inc., and is a co-founder of PreDICTA Bioscience. I.M.G.'s spouse is the CMO and an equity holder of Disc Medicine. All other authors declare no conflicts of interest.

## Inclusion and ethics

One or more of the authors of this paper self-identifies as an underrepresented ethnic and/or gender minority in science. One or more of the authors of this paper self-identifies as a member of the LGBTQIA+ community. We support inclusive, diverse, and equitable conduct of research.
