## [Transparent Peer Review file · Nature Communications]

Deeper Response Predicts Better Outcomes in High-Risk-Smoldering-Myeloma: Results of the I-PRISM Phase II Clinical Trial

Corresponding Author: Dr Irene Ghobrial

Version 0:

Reviewer comments:

Reviewer #1

(Remarks to the Author)

Review: Accept as is

The authors have satisfactorily addressed reviewer concerns.

Reviewer #2

(Remarks to the Author)

The authors addressed all of my comments raised. I do not have further suggestions to improve the manuscript, except a minor point:

1) Please cite the R packages being used. The packages list the citation information, see, e.g., <https://cran.r-project.org/web/packages/survival/index.html>, to acknowledge the programmer(s).

Reviewer #4

(Remarks to the Author)

The authors present the results of the phase II study of the combination of ixazomib, lenalidomide, and dexamethasone in patients with HR-SMM with long-term follow-up and baseline single-cell tumor and immune sequencing to help refine the population that may most benefit from early intervention. 55 patients received 9 cycles of IRd followed by 2 years of IR maintenance. The primary endpoint was PFS along with key secondary including biochemical PD and depth of response. The ORR was 93%, CR rate: 31%. After a median follow-up of 50 months, the PFS was 48.6 months and median overall survival was not reached. Whole genome or single-cell sequencing of tumor cells identified high-risk aberrations not otherwise known.

This is a well written paper and commended for all the ancillary corollary experiments performed. Have no comments

Reviewer #5

(Remarks to the Author)

The authors, Nadeem et al in this study show that high-risk smoldering multiple myeloma patients, that achieved a complete response did not experience biochemical progression or developed active myeloma (defined by SliM-CRAB criteria) after two years of therapy with Ixazomib, Lenalidomide, and dexamethasone. Achieving MRD negativity (10⁻⁵ sensitivity), also translated to a lower chance of biochemical progression.

My comments are as follows:

1. Although the study shows that early therapeutic intervention in high-risk smoldering myeloma patients might delay end-

organ damage, what the study does not help in answering is how we best define the high-risk smoldering myeloma patients that would benefit from multiple myeloma like therapy in real world clinical practice. The population that was included in the study represents a very heterogeneous smoldering myeloma cohort evidenced by the fact that patients that met any of the criteria for high-risk SMM by the different risk stratification models were classified as high risk and included in the trial. As a result as shown in Table 1b of the study, there is significant variation between the different models, with some patients that were low risk by one risk score, were high risk by another. This would make the applicability of the findings in this study difficult to apply in routine clinical practice.

2. If the objective is to achieve a deep response (CR and/or MRD negativity) with earlier intervention in high-risk SMM patients, whether treatment with a more active myeloma-like regimen with a quadruplet combination as demonstrated in the ASCENT trial (NCT03289299) instead of using a triplet would have added more strength to the study's conclusions. In addition the choice of Ixazomib as the proteasome inhibitor in the study could have reduced the effectiveness of the intervention.

3. The authors mention 11 patients experiencing biochemical progression, enrolled on other SMM-directed trials. Could this not have confounded the results for the primary endpoint, i.e., PFS for development of end-organ damage by SLiM-CRAB criteria. In addition as the authors themselves mention for patients experiencing biochemical progression the optimal management remains unclear, including the impact of earlier treatments on the future efficacy of myeloma therapy.

Minor Comments:

1. The paper would benefit with more clarity in the methods section regarding the Eligibility criteria for the study.

Reviewer #6

(Remarks to the Author)

RESPONSE TO REVIEWERS' COMMENTS

Reviewer #1 (Remarks to the Author):

Review: Accept as is

The authors have satisfactorily addressed reviewer concerns.

Response: We would like to thank Reviewer 1 for their confidence in our study.

Reviewer #2 (Remarks to the Author):

The authors addressed all of my comments raised. I do not have further suggestions to improve the manuscript, except a minor point:

1) Please cite the R packages being used. The packages list the citation information, see, e.g., <https://cran.r-project.org/web/packages/survival/index.html>, to acknowledge the programmer(s).

Response: We would like to thank Reviewer 2 for their confidence in our study. We have now included citations for the R packages that were used in the analysis described in this study. We have highlighted them in yellow in the References section.

Reviewer #4 (Remarks to the Author):

The authors present the results of the phase II study of the combination of ixazomib, lenalidomide, and dexamethasone in patients with HR-SMM with long-term follow-up and baseline single-cell tumor and immune sequencing to help refine the population that may most benefit from early intervention. 55 patients received 9 cycles of IRd followed by 2 years of IR maintenance. The primary endpoint was PFS along with key secondary including biochemical PD and depth of response. The ORR was 93%, CR rate: 31%. After a median follow-up of 50 months, the PFS was 48.6 months and median overall survival was not reached. Whole genome or single-cell sequencing of tumor cells identified high-risk aberrations not otherwise known.

This is a well written paper and commended for all the ancillary corollary experiments performed. Have no comments

Response: We would like to thank Reviewer 4 for their confidence in our study.

Reviewer #5 (Remarks to the Author):

The authors, Nadeem et al in this study show that high-risk smoldering multiple myeloma patients, that achieved a complete response did not experience biochemical progression or developed active myeloma (defined by SliM-CRAB criteria) after two years of therapy with Ixazomib, Lenalidomide, and dexamethasone. Achieving MRD negativity (10⁻⁵ sensitivity), also translated to a lower chance of biochemical progression.

My comments are as follows:

1. Although the study shows that early therapeutic intervention in high-risk smoldering myeloma patients might delay end-organ damage, what the study does not help in answering is how we best define the high-risk smoldering myeloma patients that would benefit from multiple myeloma like therapy in real world clinical practice. The population that was included in the study represents a very heterogeneous smoldering myeloma cohort evidenced by the fact that patients that met any of the criteria for high-risk SMM by the different risk stratification models were classified as high risk and included in the trial. As a result as shown in Table 1b of the study, there is significant variation between the different models, with some patients that were low risk by one risk score, were high risk by another. This would make the applicability of the findings in this study difficult to apply in routine clinical practice.

Response 1: We appreciate the comment of the Reviewer and agree that there is a lot of changes that are occurring in the risk stratification of smoldering myeloma. The study was conducted before the new IMWG criteria of 20-2-20 was developed. Our study used the criteria developed by Rajkumar (Blood 125, 3069-3075 (2015)). The same criteria were used by multiple other clinical trials including the Elotuzumab, lenalidomide and dexamethasone phase II study (Cancer Cell 40, 1358-1373.e1358 (2022) and KRD for high risk SMM (JAMA Oncol. 2021 Nov 1;7(11):1678-1685. We performed a post-hoc analysis to include the new IMWG criteria that was published after this study was already enrolled. The reason we included this table is to help with cross-study comparisons in the future. We would also like to point out that by the IMWG criteria, 46/55 patients belonged in the high and intermediate risk category (High Risk: 25, Intermediate: 21) demonstrating that a large proportion of patients would have classified as high/intermediate by the newer criteria. All other clinical trials of high-risk SMM use the Spanish or Mayo criteria or a mixture of both. Recent studies have now included the 20-2-20 criteria. This heterogeneity also reflects the real-world case where patients have varying risk scores based on different risk stratification systems. The criteria for high risk SMM may continue to evolve, this does not negate the benefit from the current responses and long term PFS data provided in this or other clinical trials.

2. If the objective is to achieve a deep response (CR and/or MRD negativity) with earlier intervention in high-risk SMM patients, whether treatment with a more active myeloma-like regimen with a quadruplet combination as demonstrated in the ASCENT trial (NCT03289299) instead of using a triplet would have added more strength to the study's conclusions. In addition the choice of Ixazomib as the proteasome inhibitor in the study could have reduced the effectiveness of the intervention.

Response: This study was conducted to assess the safety and efficacy of a 3-drug all oral regimen of Ixazomib, lenalidomide and dexamethasone at the time when triplets were the standard of care therapy for multiple myeloma. The benefits of this were to examine an all oral regimen, a proteasome inhibitor that is easier to administer and less toxic than bortezomib or carfilzomib in these asymptomatic participants, and to demonstrate whether the addition of a proteasome inhibitor can achieve a high response rate and prolonged remissions as compared to historical control len/dex data.

Other studies including ASCENT (Daratumumab, Carfilzomib, Lenalidomide and Dexamethasone) and B-PRISM (Daratumumab, Bortezomib, Lenalidomide and Dexamethasone) are indeed examining 4 drug regimens. There are many other studies asking whether another 3-drug regimen is more effective such as Daratumumab-len-dex vs. Len-dex in the ongoing phase III ECOG study or the GEM-CESAR study (carfilzomib, lenalidomide, and dexamethasone). Another study has just completed accrual using Isatuximab-len-dex vs. Len-dex (the ITHACA study).

Future studies comparing the safety and efficacy of various triplet versus quadruplet regimens could help determine the optimal treatment approach for high-risk SMM patients.

3. The authors mention 11 patients experiencing biochemical progression, enrolled on other SMM-directed trials. Could this not have confounded the results for the primary endpoint, i.e., PFS for development of end-organ damage by SLiM-CRAB criteria. In addition as the authors themselves mention for patients experiencing biochemical progression the optimal management remains unclear, including the impact of earlier treatments on the future efficacy of myeloma therapy.

Response: Thank you for your comment. The last follow-up for these 11 patients is recorded at the time they discontinue the study treatment, which helps limit confounding as these patients are censored after that. For patients progressing by SLiM-CRAB criteria, we observed biochemical progression as a preceding or concurrent event with end-organ damage. Therefore, any biochemical progression observed does not independently confound the progression-free survival (PFS) endpoint defined by SLiM-CRAB criteria.

Minor Comments:

1. The paper would benefit with more clarity in the methods section regarding the Eligibility criteria for the study.

Response: We previously included the eligibility criteria in the Supplemental Materials to meet the 3000-word limit of submission but we have now included the eligibility criteria in the Methods section of the manuscript.

Reviewer #6 (Remarks to the Author):

I co-reviewed this manuscript with one of the reviewers who provided the listed reports. This is part of the NatureCommunications initiative to facilitate training in peer review and to provide appropriate recognition for Early Career Researchers who co-review manuscripts.

Response. We thank you for your review.